# Adaptive Downward/Upward Routing Protocol for Mobile-Sensor Networks

**Jinpeng Wang, Gérard Chalhoub ***[ID] **and Michel Misson**

LIMOS/CNRS, University of Clermont-Auvergne, 63170 Aubière, France; jinpeng.wang@uca.fr (J.W.);
michel.misson@uca.fr (M.M.)

\* Correspondence: gerard.chalhoub@uca.fr

**Abstract:** Recently, mobility support has become an important requirement in various Wireless Sensor Networks (WSNs). Low-power and Lossy Networks (LLNs) are a special type of WSNs that tolerate a certain degree of packet loss. However, due to the strict resource constraints in the computation, energy, and memory of LLNs, most routing protocols only support static network topologies. Data collection and data dissemination are two basic traffic modes in LLNs. Unlike data collection, data dissemination is less investigated in LLNs. There are two sorts of data-dissemination methods: point-to-multipoint and point-to-point. In this paper, we focus on the point-to-point method, which requires the source node to build routes to reach the destination node. We propose an adaptive routing protocol that integrates together point-to-point traffic and data-collection traffic, and supports highly mobile scenarios. This protocol quickly reacts to the movement of nodes to make faster decisions for the next-hop selection in data collection and dynamically build routes for point-to-point traffic. Results obtained through simulation show that our work outperforms two generic ad hoc routing protocols AODV and flooding on different performance metrics. Results also show the efficiency of our work in highly mobile scenarios with multiple traffic patterns.

**Keywords:** LLN; WSN; routing; mobility

## 1. Introduction

Wireless Sensor Networks (WSNs) have been widely developed in the last decade, and new applications are emerging rapidly. Most of these applications are used for surveillance and recording environmental or physical conditions [1]. In these applications, data collection is the basic traffic mode, where all traffic in the network is destined to a predefined destination called the sink node. We use the word upward to describe the transmission direction of packets from sensor nodes towards the sink. Thus data collection is also called upward routing in this paper. If we consider the Internet of Things (IoT) application domains, upward routing is not the only traffic mode in the network. The sink node needs to send commands to certain sensor or actuator nodes to perform actions according to received information. In these applications, another traffic mode coexists with the upward routing which is from the sink node to certain sensor nodes, which we call downward routing.

Compared with upward routing, downward routing is much less studied in WSNs. The major downstream methods use flooding as the basic dissemination method, which propagates data or control messages to the entire network. This operation costs a lot of resources (energy, bandwidth, computation, etc.) during transmission and causes network congestion. Some routing protocols, like Routing Protocol for Low-Power and Lossy Networks (RPL), support downward routing by unicasting data to the destination through a route path embedded in data-packet headers [2]. Our proposal is based on a similar technique.

Nowadays, mobility is an important requirement in many applications. In a monitoring application, if the observed object is mobile, the network topology is dynamic. In our previous work [3], we have already proven that traditional upward routing protocols do not support mobility well. Due to the movement of nodes, topology continuously changes and links between nodes are unstable. Nodes need to quickly react to adapt to the movement before making a decision for the next-hop. It is also a hard task for downward routing to cope with mobility. Each time the topology changes, most of the paths to reach every node by the sink are different. Thus, the sink needs to quickly reconstruct the topology and rebuild the routes to reach every node in the network, or data are delayed or lost.

In this paper, we concentrate on mobile scenarios where all nodes except the sink are free to move, and need to send periodic data to the sink. Simultaneously, the sink needs to periodically send command packets to randomly selected nodes in the network. We propose a routing protocol, Adaptive Downward/Upward Protocol (ADUP), which supports both upward and downward routing. Mobility support of upward routing is achieved according to RRD+ (RSSI, Rank and Dynamic) [4].

The remainder of the paper is organized as follows. In Section 2, we present related work concerning downward routing in WSNs. Section 3 describes our contribution: the ADUP protocol. In Section 4, we present and analyze simulation results that show the efficiency of ADUP. Finally, we conclude the paper and give some future investigations in Section 5.

## 2. Related Work

Downward routing is less studied than upward routing in WSNs. It can be mainly classified into three categories: broadcast-based, unicast-based, and broadcast- and unicast-based. Most of these protocols are proposed for static scenarios and only few of them can cope with mobile scenarios.

Glossy [5] and LWB [6] are broadcast-based routing protocols that use flooding as the basic dissemination method for propagating data to the entire network. Ferrari et al. [5] proposed a flooding and time synchronized protocol for WSNs called Glossy. Glossy exploits the flooding mechanism to implicitly synchronize the network. According to clock values embedded in flooding packets, all receivers synchronize relatively to the clock of the initiator. Each packet also embeds a relay counter value, which represents how many times a packet has been relayed. Nodes always concurrently transmit packets with the same relay counter. Benefiting from concurrent transmission and synchronization, Glossy avoids interference during flooding and profit from constructive interference. However, Glossy highly relies on concurrent transmissions and actuation of initiators, and this results in Glossy not supporting data-collection scenarios, since too many initiators cause serious congestion in the network and congestion induces inaccurate synchronization. In order to obtain precise synchronization, Glossy also has a very strict packet-size limit during transmission and this makes the propagation of variable-sized packets impossible. Low-Power Wireless Bus (LWB) is an updated version of Glossy, which concurrently supports one-to-many, many-to-one, and many-to-many traffic. Unlike Glossy, which is simply driven by the events of initiators, LWB appoints a node in the network to work as a controller and it sends schedules of each initiator. Although LWB reduces congestion using a central control method, it still meets the same problems of a limited number of initiators and packet sizes as Glossy. In addition, neither Glossy nor LWB supports mobility.

RPL is a unicast-based routing protocol for Lossy and Low Power Networks (LLNs) that supports upward and downward traffic patterns. RPL supports two modes of downward traffic: Nonstoring mode and Storing mode. In Nonstoring mode, Destination Advertisement Object (DAO) messages are directly sent to the Destination-Oriented Directed Acyclic Graph (DODAG) root along a default route. The root establishes source-routing table entries for destinations learned from DAOs. Before sending a data packet, the root uses source routing to completely specify the route of the packet. In Storing mode, DAOs are sent to all parent nodes to inform them of the existence of a child node. Once a data packet is sent by the DODAG root, it must be sent to all one-hop neighbors first. Afterward, the data packet is sent hop by hop until it reaches its destination or hop limit. In our previous work [4],



we already showed that RPL does not support upward routing well in mobility. It is also hard for RPL to support downward routing well in mobility. Due to topology changes in mobility, parent nodes change frequently. This requires RPL to send DAOs on time in order to keep the downward routing table up to date, which costs too much overhead. Otherwise, packets are lost due to outdated path information. Carels et al. [7] proposed a new mechanism to improve RPL downward route updating of Nonstoring mode in mobility. Instead of sending a no-path DAO to a mobile parent node, a child node sends a DAO containing no-path information to a static parent node to reach root. However, this method suffers from high overhead and it relies on the presence of fixed nodes. This means that this method cannot operate in a scenario where all nodes are mobile.

Due to the inefficiency of DAOs in downward routing, Duquennoy et al. [8] proposed an opportunistic routing protocol called Opportunistic RPL (ORPL), which is built on-top of RPL. ORPL deactivates DAO and simply uses the broadcasting of DODAG Information Object (DIO). Each node owns a routing set that is a set of nodes with lower ranks and this set is propagated inside DIO messages. The routing set allows nodes to know whether a node is on a path to the destination or not. ORPL uses anycast instead of unicast to propagate a data packet. Nodes that receive the packet decide whether to forward it or not. If the destination of a packet is in the routing set, this packet continues to be relayed; otherwise, the packet is dropped. ORPL is proposed for static scenarios only. When considering mobility, rank and routing sets need to be updated on time according to the movement of nodes. Moreover, due to the fact that the updating of routing sets depends on the propagation of DIOs, timely updating costs too much overhead and is not practical.

Dynamic Source Routing (DSR) and Ad hoc On-Demand Distance Vector (AODV) are unicast-based downward routing protocols. They both employ flooding methods to support route discovery and route maintenance. Compared with DSR, AODV further supports periodic advertisements and distance vector routing, which is more adaptive to dynamic scenarios. However, both DSR and AODV need to run route discovery and route maintenance very often in order to update routing tables in a timely manner in mobile scenarios. In this process, the flooding of requests causes congestion in the network, and route discovery and maintenance cannot run well. Improving AODV based on restricted broadcasting is a common method proposed in References [9,10] for vehicular networks. However, these methods use geographic positions to assist broadcasting reduce the number of retransmissions, and thus require each node to be equipped with a Global Positioning System transceiver, which is difficult to achieve in WSNs and LLNs.

Opportunistic source routing (OSR) [11] is a broadcast- and unicast-based downward routing protocol that introduces opportunistic routing into traditional source routing. OSR uses a bloom-filter mechanism to encode a downward source route, which reduces the length of a packet header while processing source routing. OSR uses multiple traffic-flow patterns, unicast, multicast, and broadcast. Unicast and multicast are the main ways to propagate information. In the case of failure of unicast and multicast, broadcast is used. OSR achieves a reduction in transmission count and a gain in reliability compared to standard RPL. However, OSR also needs route discovery and maintenance to deal with mobile scenarios, which meets congestion problems, similarly to DSR and AODV.

## 3. Adaptive Downward/Upward Protocol

In this paper, we consider upward and downward routing in mobile scenarios. In these scenarios, we suppose that all nodes are free to move except the sink node. In upward routing, every node periodically generates data that are destined to the sink at a constant rate. Before sending packets, each node selects a next-hop from its neighbors based on RRD+. RRD+ helps to cope with the frequent-topology-changes problem in mobility. In downward routing, the sink periodically sends command packets to nodes. Due to the movement of nodes, routes from the sink to sensor nodes do not stay the same. If route information cannot be updated on time, packets may be lost. In what follows, we present our method, which is an extension of RRD+ that copes with downward routing in mobility.

### 3.1. Overview of RRD+ Mechanism

We integrated downward routing in RRD+, which was originally designed for upward routing. RRD+ is a routing mechanism that can be used by hierarchical routing protocols to cope with mobility in convergecast data-collection scenarios. It is based on link quality monitoring and Rank value updating to better adapt to movement, and makes fast decisions on selecting next-hop neighbors. Moreover, RRD+ supports a dynamic management of control messages in order to reduce the overhead in the network. In what follows, we describe the different aspects of RRD+.

#### 3.1.1. Movement Direction Monitoring

RRD+ uses variation of Received Signal Strength Indicator (RSSI) to monitor movement direction. Nodes obtain RSSI values from acknowledgement (ACK) messages and control messages. A node manages two RSSI values for each parent node, Old RSSI value and New RSSI value. Old RSSI is retrieved from the previous ACK or control message, and New RSSI value is obtained from currently received ACK or control message. According to the variation of New RSSI value with regards to Old RSSI value, RRD+ estimates and monitors the movement direction of nodes. When a New RSSI value is lower than an Old RSSI value, RRD+ considers that the node is moving away from its parent node. Otherwise, RRD+ considers that the node is moving closer to its parent node.

#### 3.1.2. Link-Quality Monitoring

Due to unpredictable path attenuations, RSSI values might vary even when neither node moves. In order to take this phenomenon into account, we introduce two RSSI thresholds: Safety Threshold and Hysteresis Threshold, where the Safety Threshold is larger than the Hysteresis Threshold, as shown in Figure 1. We used a dotted line for Safety Threshold, and a dashed line for Hysteresis Threshold. Note that, due to the nature of wireless-signal propagation, in reality both RSSI thresholds and transmission range are most likely to look like a cloud and in our simulation model we used a probabilistic propagation model to take into account coverage-zone instability.

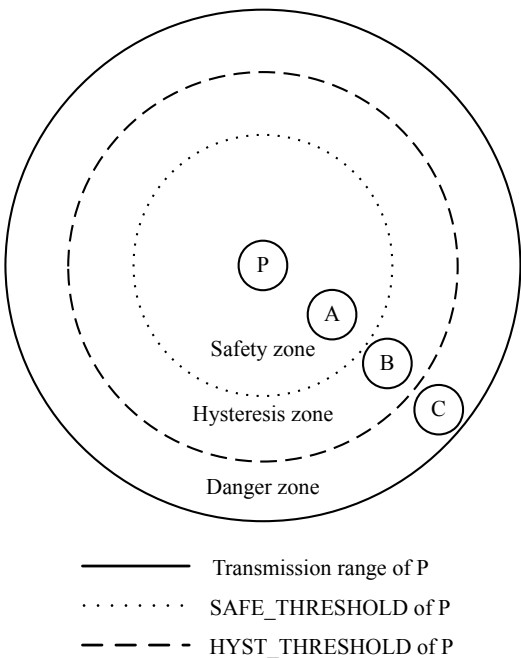

**Figure 1.** Node P is the parent node of nodes A, B, and C. A is in the Safety zone of node P. B is in the Hysteresis zone of node P. C is in the Danger zone of node P.

When New RSSI is higher than or equal to Safety Threshold, the node is considered to be in the Safety zone of its parent node, and it has good link quality with it; this is the case of node A in Figure 1. When New RSSI is smaller than Safety Threshold but higher than Hysteresis Threshold, which is the case of node B in Figure 1, we need to first detect movement direction and then consider whether to stop using the link or not. In order to reduce coverage-zone variation influence, we add a hysteresis value to Old RSSI when comparing it to New RSSI. When New RSSI is smaller than Hysteresis Threshold, which is the case of node C in Figure 1, only New RSSI and Old RSSI are used to estimate direction without using hysteresis.

### 3.1.3. Rank Updating

Rank mechanism, which is proposed by RPL, is also an important part of RRD+. The Rank of a node is a value that defines the position of the node with respect to the sink in terms of routing metrics. The Rank of the sink node is *ROOT_RANK* and *MinHopRankIncrease* is the minimum increase of the Rank between a node and any of its parent nodes. The rank value is proportional to the increase of the metric contained in control messages; therefore, the Rank of a node is calculated as shown in Equation (1).

$$Rank = ROOT\_RANK + axMinHopRankIncrease \tag{1}$$

where *a* is a value included in control messages that come from lower rank nodes.

A node is not allowed to send data packets to neighbors with higher or equal Ranks, which is an effective way to avoid loops in the network. In mobility scenarios, the position of a node frequently changes. The original Rank mechanism does not offer methods to update the Rank in a timely manner. This causes loops when a parent node becomes a descendant node. RRD+ monitors link existence and movement direction to allow nodes to update their Ranks in a timely manner. The goal is to update the Rank of a node when it is about to lose its link with its current parent node based on the link quality-monitoring mechanism.

### 3.1.4. Dynamic Control Message Management

In mobile scenarios, propagation of control messages needs to be more frequent in order to adapt to topology changes. Maintaining up-to-date information about topology causes high overhead. In our case, similarly to RPL, control messages are broadcast by the sink node and propagated by other nodes until they reach leaf nodes. In RRD+, we designed a dynamic control message management according to Rank values in order to reduce overhead. Nodes that are closer to the sink should send control messages more frequently and the frequency is reduced for nodes with higher Ranks. Control messages coming from lower Rank nodes will help more nodes find parent nodes. When a node changes its Rank value, it automatically adapts its control message interval. The control message interval calculation is shown in Equation (2).

$$Interval = Base\_interval + RankxTime\_unit \tag{2}$$

where *Interval* dynamically changes due to the change of Rank of nodes in mobility. *Base_interval* is the smallest *Interval*. *Rank* stands for the current Rank value of the node. *Time_unit* is the incremental step in the control message frequency. *Base_interval* and *Time_unit* can be fixed according to the application needs. High densities and high speeds would require smaller values of Interval.

### *3.2. Dynamic Next-Hop Table*

In RRD+, all neighbors with lower Ranks form a set that we call a parent set. In a data-collection process, before sending a packet, a node needs to select a next-hop from its parents set. In ADUP, the ID of this next-hop is included into data packets and sent to the sink. Instead of storing the entire addresses of next-hop nodes in data packets, we only use 1 byte to store the ID of the next-hop node of the source node. Due to the fact that the upper limit value of 1 byte is 255, the maximum number of

nodes in the network cannot exceed 255. Figure 2 shows the fields of upward data packets. When the sink receives data packets, it builds a next-hop table as shown in Figure 3. The first column of this table stands for the ID of nodes, except the sink, in the network. We consider that there are $n$ nodes and one sink in the topology. We define the ID of each node as $ID_i \in \{ID_0, ID_1, ID_2, ..., ID_n\}$, where $ID_0$ stands for the ID of sink. The second column stands for the ID of best next-hop for each node referred to as $N(ID_i)$. Note that $N(X) \in \{ID_0, ID_1, ID_2, ..., ID_n\}$ and $X \in \{ID_1, ID_2, ..., ID_n\}$.

| 128 Bytes | | | | | |
|---|---|---|---|---|---|
| 0-29 | 30-31 | 32-33 | 34 | 35-36 | 37-127 |
| Data | Sender address | Receiver address | ID of next-hop | Packet Id | Idle |

**Figure 2.** Fields of upward data packets.

| ID | Nexthop |
|---|---|
| $ID_1$ | $N(ID_1)$ |
| $ID_2$ | $N(ID_2)$ |
| ... | ... |
| $ID_n$ | $N(ID_n)$ |

**Figure 3.** Dynamic next-hop table of the sink node.

RRD+ updates the Rank value of each node according to link quality and movement direction. Nodes in a parents set are automatically removed or added based on the variation of Rank values. Thus, the next-hop of each node also dynamically changes according to movement. Due to the fact that each node periodically sends packets to the sink, the next-hop table periodically adapts to mobility.

*3.3. Route Building in Downward Routing*

In order to reach the destination through multiple hops, the sink needs to build a route before sending a packet. Algorithm 1 depicts the route-building process. We use $ID_d$ to stand for the ID of the destination node which is put into the route first. The sink extracts the preferred next-hop of $ID_d$ from the dynamic next-hop table. If $N(ID_d)$ equals $ID_0$, this means that the sink can directly reach node $ID_d$, and the building process immediately stops. In case $N(ID_d)$ is not $ID_0$, the sink continues the route-building process. For any entry in the first column of dynamic next-hop table, the entry $ID_i$ that equals $N(ID_d)$ is put into the route, and its next-hop $N(ID_i)$ needs to be compared to $ID_0$. The building process stops when the next-hop of item $ID_i$ is $ID_0$. The building process of a route is done from the destination to the sink, the route we get is in reverse order, and it is thus reversed before it is used as a path.

Figure 4 shows how Algorithm 1 works. $ID_d$ is put into the route first. During this process, the numbers of $ID_i$ are put in the route until the next-hop of $ID_x$ is found to be $ID_0$. At the end, the route needs to be reversed.

We consider that there are $m$ nodes in the route. Before sending a data packet, the sink needs to store the IDs of these nodes in the data packet as shown in Figure 5. Every time the data packet is relayed, the route offset will be increased to help relay nodes to find the next-hop within $m$ bytes until reaching the destination.

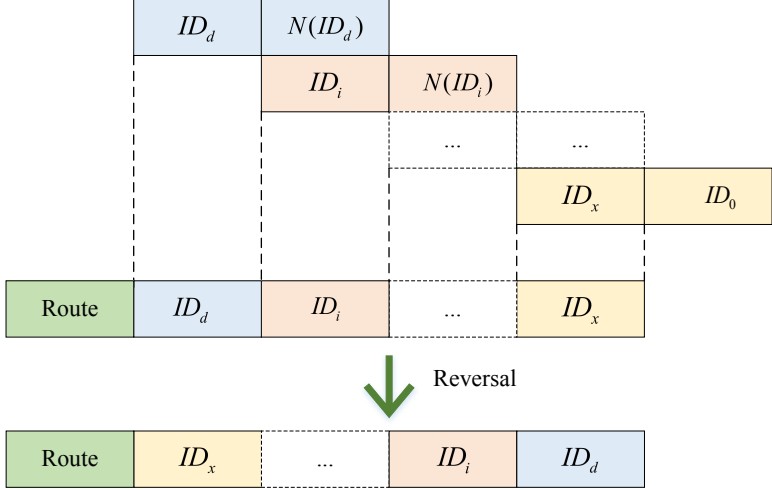

**Figure 4.** Route-building process.

---

**Algorithm 1:** Route building.

---

**Input:** $ID_d$
**Output:** *Route*
**begin**
   Put $ID_d$ in the *Route*;
   $Nexthop = N(ID_d)$;
   **while** *Nexthop does not equal to* $ID_0$ **do**
      **for** *each item i in* $ID_i$ **do**
         **if** $ID_i = Nexthop$ **then**
            $Nexthop = N(ID_i)$;
            Put $ID_i$ in the *Route*;
         **end**
      **end**
   **end**
   Reverse(*Route*);
**end**

---

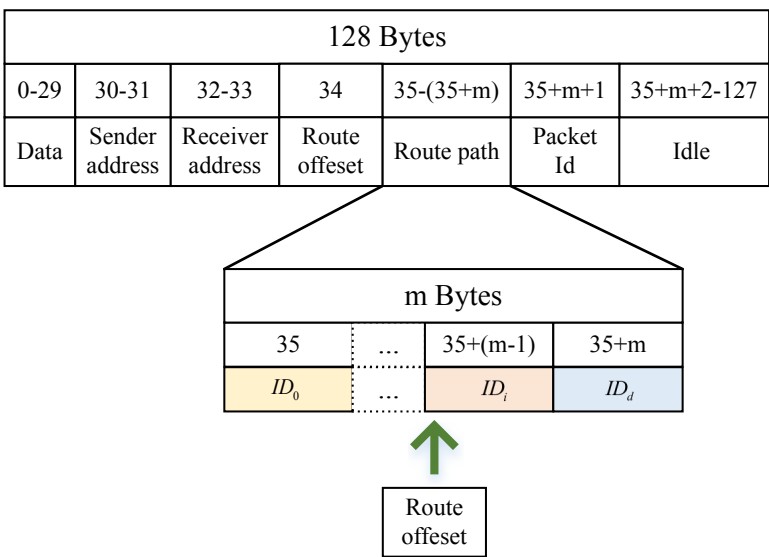

**Figure 5.** Fields of downward data packets.

## 4. Simulation Environment and Performance Evaluation

In this section, we describe our simulation setup and present the evaluation results.

### 4.1. Simulations Parameters

We evaluated the performance of ADUP by doing simulations using the Cooja simulator [12]. There are four propagation models in Cooja [13], and we used Multipath Ray-tracer Medium (MRM), which simulates real environments with realistic unstable link qualities. In order to make it more suitable for urban and unstable environments, we included random behavior to path-loss calculation of MRM. Indeed, we added a Gaussian random variable in the path-loss formula to simulate the instability of the radio links. We calibrated the randomness to make transmission range randomly fluctuate between 30 and 50 m, independently for each transmission.

At the beginning of the simulation, all nodes are randomly deployed within a 200 × 200 m area, and they are free to move within this area. Every 5 s, velocity is randomly chosen from [1, 3 m/s], and the direction of nodes changes by choosing a random destination position inside the deployment area. Under this mobility model, we set the minimum control message interval as 2 s according to the experimental test, which means the minimum topology update interval is 2 s. In order to make sure that ID of next-hop updates can be sent to the sink on time, the upward transmission interval should be shorter than 2 s. We set the upward transmission rate to 1 pkt/s. Every 1 s, all nodes except the sink generate a data packet to the sink. Meanwhile, the sink generates command packets to a randomly chosen node in the network every 1 s. Every scenario simulates 5 min of network activity. Table 1 summarizes the simulation settings.

**Table 1.** Simulation setup.

| Item | Parameters |
| --- | --- |
| Network simulator | Cooja under contiki OS (3.0) |
| Radio propagation model | MRM with random behavior |
| Medium access control | CSMA/CA |
| Simulation time | 5 min |
| Emulated platform | Sky starter platform |
| Sensor Nodes Deployment | Random Deployment |
| Data size | 30 Bytes |
| Packet queue size | 16 |
| Upward transmission rate | 1 pkt/s |
| Downward transmission rate | 1 pkt/s |
| Transmission power | −20 dBm |
| Transmission range | [30 m, 50 m] |
| Number of nodes | 20, 40, 60 |
| Area of deployment | 200 m × 200 m |
| Frequency range | 2.4 GHz |
| Mobility model | Random Waypoint |
| Minimum speed | 1 m/s |
| Maximum speed | 3 m/s |
| Speed-changing interval | 5 s |
| New location pause time | 5 s |

### 4.2. Simulation Results

In order to assess the efficiency of ADUP in dealing with mobility, we compared it to two other existing protocols that cope with mobility: AODV and Flooding. In addition, AODV and Flooding are generic protocols that are designed for any application scenario containing upstream and downstream traffic. As we introduced in Section 2, AODV is a typical unicast-based routing protocol. It uses route discovery and route maintenance to support dynamic topologies. Furthermore, it is based on periodic advertisements and distance vector routing, which is more adaptive to mobile scenarios compared

to DSR for example. Flooding is a broadcast-based routing protocol that is not sensitive to mobility. It may support mobile scenarios well with few transmission events. It is also interesting to estimate the performance of Flooding in mobile scenarios with different traffic patterns. In order to limit the number of transmissions using Flooding routing protocol, nodes only route the same packet once. We used unique identifiers for packets in order to manage this issue.

Note that in this paper we did not compare ADUP to RPL because in previous papers we showed that RPL does not cope well with mobility, and that RRD+ outperforms RPL.

We used four performance metrics to evaluate the efficiency of these protocols: (i) packet-delivery ratio, (ii) average end-to-end delay, (iii) dropped packet ratio, and (iv) number of control packets. For each network size, we generated 10 different random mobility scenarios. Each performance metric was averaged over 10 iterations for each network size.

4.2.1. Packet-Delivery Ratio

Packet delivery ratio is the ratio between the number of received data packets at the destination and the number of generated packets by the source nodes. Figure 6 shows packet delivery ratio based on different traffic patterns.

Figure 6a shows that ADUP outperforms AODV and Flooding in terms of packet-delivery ratio for upstream traffic. ADUP has about 10% improvement over AODV. This is mainly due to the fact that, in upward routing, ADUP only needs to update next-hops rather than whole path routes, which helps adapt faster to mobility. Moreover, ADUP and AODV are much better than Flooding because it uses broadcasting as a basic traffic pattern, which causes serious network congestion, especially when the number of senders is high.

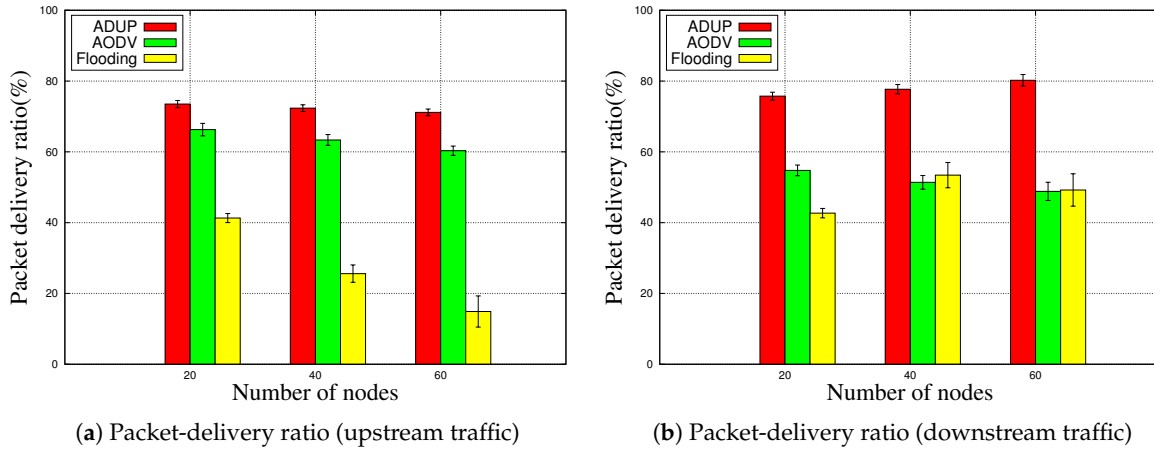

(**a**) Packet-delivery ratio (upstream traffic)　　　　(**b**) Packet-delivery ratio (downstream traffic)

**Figure 6.** Packet delivery ratio.

Figure 6b shows that ADUP is much better than AODV and Flooding for downstream traffic. This is mainly due to the fact that benefiting from upstream traffic ADUP allows the sink to update the path to reach any node in the network in a timely manner. Results also show that the packet-delivery ratio of AODV decreases when the number of nodes increases. The reason is that, with more nodes, upstream traffic is higher, which causes more congestion and a higher risk of collisions. Affected by upstream traffic, the downstream traffic of AODV would lose more packets during transmission. However, unlike AODV, the packet-delivery ratio of Flooding on downstream traffic first increases for 40 nodes, and then decreases for 60 nodes. This is mainly due to the fact that Flooding uses broadcasting rather than unicasting. When the number of nodes increases, broadcasting would help the same packet being relayed more times, which would increase the opportunity of successfully sending a packet to reach the destination. This allows Flooding to increase its packet-delivery ratio of downstream traffic with 40 nodes. However, the downstream traffic of Flooding is also affected by the

upstream traffic. When there is serious network congestion caused by the upstream traffic of 60 nodes, the packet-delivery ratio of downstream traffic decreases.

### 4.2.2. Average End-To-End Delay

End-to-end delay is the duration between the instant the packet is generated by a node and the instant this packet is received by the sink node. Figure 7 shows the average end-to-end delay based on different traffic patterns. The average end-to-end delay is calculated over the total received packets only. This means that these results do not include the end-to-end delay of packets that are lost in the network.

Figure 7a shows that ADUP outperforms AODV on average end-to-end delay for upstream traffic. This is mainly due to the fact that ADUP can quickly adapt to mobility and dynamically update parents set according to the movement of nodes, which reduces time interval before sending a packet. However, AODV needs to build routes before transmission. This process is more frequent in mobility, since link failure happens more often, which results in longer delays before sending a packet with AODV. Compared with ADUP and AODV, the average end-to-end delay of Flooding increases exponentially when the number of nodes increases. The reason is that Flooding broadcasts packets to the entire network which causes network congestion. Congestion causes packets to be delayed at each hop until they reach the sink. It is worth noting that, although ADUP and AODV achieve similar average end-to-end, ADUP was able to reach that delay for a higher packet-delivery ratio.

Figure 7b shows that the average end-to-end delay of ADUP outperforms that of AODV and Flooding for downstream traffic. This is mainly due to the fact that upstream traffic helps ADUP quickly update route information. Packets sent by the sink are relayed to the destination in a timely manner. The same as upstream traffic, the average end-to-end delay of Flooding on downstream increases exponentially as well. Network congestion caused by upstream traffic is also the reason.

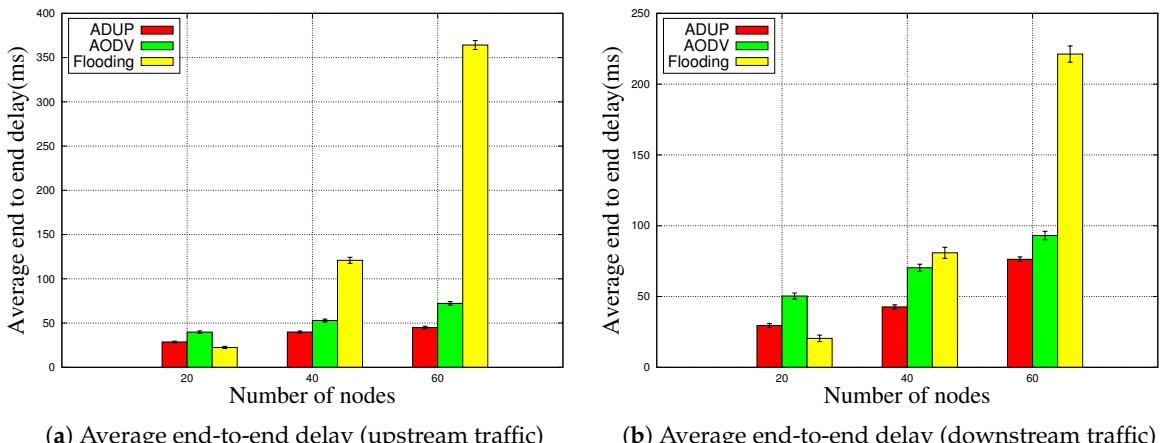

(**a**) Average end-to-end delay (upstream traffic)    (**b**) Average end-to-end delay (downstream traffic)

**Figure 7.** Average end-to-end delay.

### 4.2.3. Dropped Packets Ratio

The number of dropped packets is the number of packets that are dropped after exceeding the maximum number of retransmission attempts. Dropped-packets ratio is the ratio between the number of dropped packets and the number of generated packets. Figure 8 shows the dropped-packets ratio for different traffic patterns. Due to the fact that Flooding directly broadcasts packets, there are no retransmissions and thus no dropped packets. Hence, in Figure 8 we only show a comparison between ADUP and AODV.

Figure 8a shows that ADUP outperforms AODV in terms of dropped packets ratio for upstream traffic. This is mainly due to the fact that link quality monitoring and rank updating helps ADUP detect movement in advance and select next-hop nodes with good link quality. This increases the success

rate of sending a packet and reduces the number of dropped packets. Results also show that dropped packets ratio of ADUP and AODV for upstream traffic first increases with 40 nodes and then decreases with 60 nodes. When the number of nodes increase from 20 to 40, there is more upstream traffic, which causes more collisions and retransmissions. Thus, the dropped-packets ratio increases with 40 nodes. However, when the number of nodes continues to increase, network density becomes higher. A node would have more available next-hops to select from. This helps nodes to select next-hops with better link quality, which results in fewer dropped packets.

Figure 8b shows that ADUP also outperforms AODV in terms of dropped packets for downstream traffic, and the ratio of ADUP even decreases when the number of nodes increases. Benefiting from upstream traffic and route building in downward routing, ADUP helps the sink node build downward routes to reach any node in the network. When the number of nodes increases, the density of nodes increases. Next-hop nodes with better link quality would be used by the sink, and this would enhance the path quality between the sink and the destinations. This also helps reducing the number of dropped packets. However, unlike ADUP, the dropped-packets ratio of AODV for downstream traffic increases slightly when the number of nodes increases. This is mainly due to the fact that AODV is a reactive protocol and it builds routes only when there is data to be transmitted. This process delays transmission, which results in AODV not being able to update topology in a timely manner. This causes packet loss due to inappropriate next-hop selection.

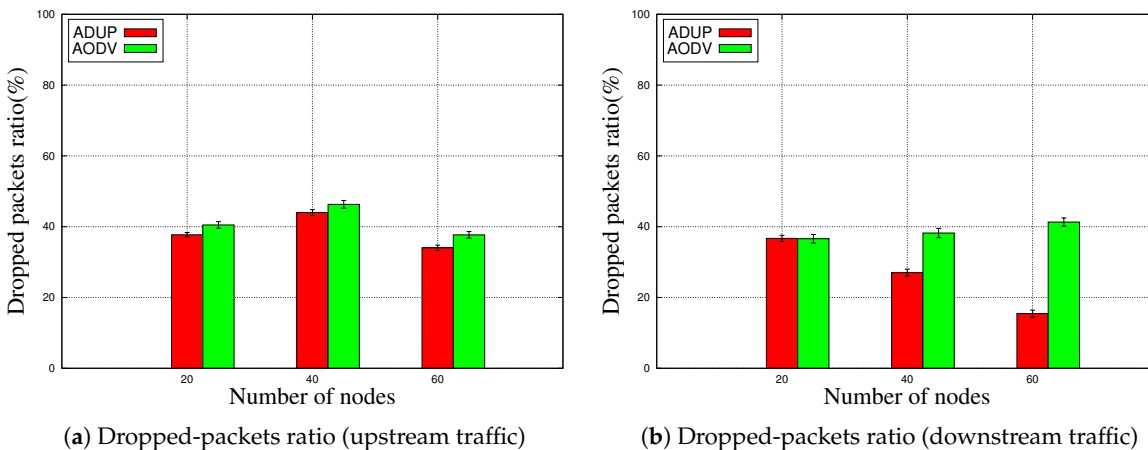

(**a**) Dropped-packets ratio (upstream traffic)　　　　　(**b**) Dropped-packets ratio (downstream traffic)

**Figure 8.** Dropped-packets ratio.

### 4.2.4. Number of Control Packets

The number of control packets is the sum of control packets that are sent during the simulation by the routing protocol. ADUP only contains one type of control message that is broadcast by the sink node and propagated by other nodes until it reaches the leaf nodes. AODV generates three types of control messages: Route Request (RREQ) messages, Route Reply (RREP) messages, and Route Error (RERR) messages. RREQ is used in route-discovery processes in order to build a route to reach the destination node. RREP is sent once the destination node receives a RREQ or an intermediate node has an active route to the destination. RERR is sent whenever a node detects a link failure or does not have an active route to the destination. Flooding only uses broadcasting as traffic patten, and it does not need any additional control packets. Thus, in Figure 9 we only show a comparison between ADUP and AODV.

Figure 9 shows that ADUP has very low overhead compared to AODV, especially when the number of nodes increases. This is due to two reasons. First, ADUP uses dynamic control message management to reduce the number of control packets used by upstream traffic, which helps to save overhead for upstream traffic. Second, ADUP embeds next-hop information in data packets destined for the sink. This allows the sink to build routes for downstream traffic without generating additional

control traffic. Moreover, AODV broadcasts RREQ to ask for a route, which adds more overhead to the network, especially when the number of nodes increases.

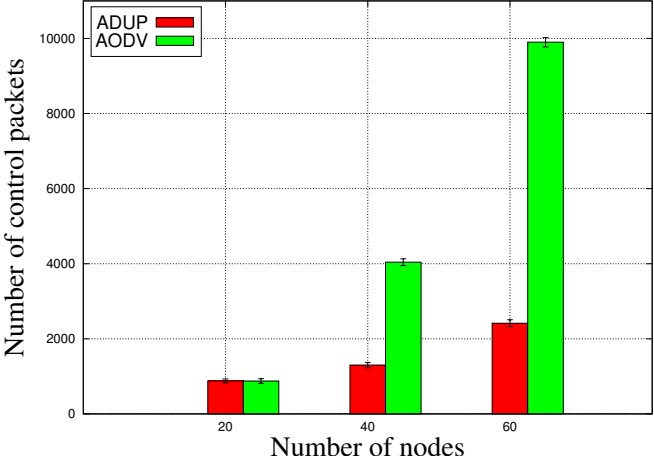

**Figure 9.** Number of control packets.

## 5. Conclusions

In this paper, we proposed ADUP, a routing protocol that concurrently supports upward and downward routing in mobile scenarios. It is suitable for application where all nodes are mobile and send periodical data packets to a control station that is called the sink. In addition, this sink needs to periodically contact other nodes of the network. This is the case for many monitoring and surveillance applications.

The support of upward routing in mobility is based on our previous work, RRD+, which copes well with mobility and helps nodes make decisions on the selection of next-hops. The support of downward routing in mobility is extension work on RRD+. Nodes embed the best next-hop of nodes in the packet headers and send it to the sink inside the periodical data packets. Once the sink receives the data packets, it builds a next-hop table. Based on this table, the sink is able to build the route to reach any node in the network. Due to the fact that the best next-hop is periodically selected from a dynamic parent set, the next-hop table is also dynamic and could adapt to mobility in a timely manner. The end result for downward routing is low on overhead because the only overhead is from the embedded information in the headers of collected data packets.

We implemented our work in Cooja and compared it with two other generic routing protocols, AODV and Flooding. Results show that ADUP outperformed AODV and Flooding on different performance metrics in mobility scenarios. Results also show that ADUP simultaneously supports upward and downward routing in dense and highly mobile scenarios. In the upward-routing process, ADUP helps nodes detect next-hops with good link quality. In the downward-routing process, ADUP helps the sink node build routes to reach any node in the network (given that this node is generating traffic towards the sink).

In our future work, we plan to implement ADUP on a testbed or in a real outdoor scenario in order to test its efficiency when faced with real-life radio-link conditions. Even though we included link instability in our simulator in order to emulate real-life links, in a real scenario nodes also suffer from interference coming from other nearby active technologies. Emulating interference in the simulator is a difficult challenge and it will be interesting to see how ADUP reacts to this in real life. Moreover, in order to support large-scale networks with hundreds of nodes, it is necessary to introduce a similar mechanism to bloom filter in order to encode downward routes in ADUP, which could reduce the length of packet headers. In addition, we plan on supporting duty cycling and evaluating the impact of energy saving on network performance.

**Author Contributions:** Conceptualization, J.W. and G.C.; methodology, J.W. and G.C.; software, J.W.; validation, J.W. and G.C.; investigation, J.W. and G.C.; resources, J.W. and G.C.; data curation, J.W.; writing—original draft preparation, J.W. and G.C.; writing—review and editing, J.W. and G.C.; visualization, J.W.; supervision, G.C.; project administration, G.C. and M.M.; funding acquisition, G.C. and M.M.

**Funding:** This research was conducted with the support of the European Regional Development Funds (FEDER) program of 2014–2020 and the regional council of Auvergne.

**Conflicts of Interest:** The authors declare no conflict of interest.

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
