# Peer review of "Adaptive Downward/Upward Routing Protocol for Mobile-Sensor Networks"

_futureinternet, doi:10.3390/fi11010018_

Round 1

Reviewer 1 Report

The paper presents an extension to a routing protocol for low-power and lossy networks that is able to transmit traffic from a sink to a set of mobile nodes in an efficient manner. The text is, in general, well presented, with an adequate English language, although there exists some little mistakes here and there, and a nice structure. However, I feel that the proposed extension is straightforward and not completely studied. So I cannot recommend the paper for publication in its current form.

In particular, I would like to know more details about the transmission of the of the next-hops from the leaf nodes to the sink, for instance: what is the overhead to transmit that list, and how that would limit the size of the network (both number of nodes and physical size as it is related to the number of ranks and so, hops). Would it be possible to implement any kind of compression or to take advantage of the fact that not all node-parent relationships change simultaneously.

In a similar vein, has the possibility to distribute the routing information across the network been considered? Would it be possible or adequate?

In the experimental section I miss some discussion about the relationship between the data transmission period by the sink and the combined speed and density of the nodes. Clearly with higher densities or higher speed the topology of the network changes more rapidly and the algorithm adaptation speed depends inherently on the transmission period. It would be would to now what are the fundamental limits of the algorithm, even it is only determined experimentally.

Finally, it would be would to provide a comparison with [10].

Minor comments:
* Page 1, line 30: (…) which propagates data (…).
* Page 1, line 31: (…) a lot of resources (…)
* Page 1, line 33: Please provide a reference for RPL
* Page 2, line 67: relies highly
* Page 2, line 74: will take change of sending
* Page 2, line 80: Please define both DAO and DODAG
* Page 3, line 131: will help to cope with the frequent
* Page 4, line 144: estimate link quality
* Page 5, line 179: We consider that there are
* Page 5, line 207: Please, do not use * as the multiplication symbol.
* Page 6, algorithm 1: Shouldn't the while condition be "while Nexthop is not equal to ID_0"?
* Page 7, line 237: present->percent

Author Response

We would like to thank the reviewer for their work. Please find in the attached file our answers to each of their comments and requests.

Reviewer 2 Report

The authors present a protocol to improve the performance of wireless networks similar to those of sensors, in which information is transmitted bidirectionally. The authors compare their proposal with two well-known algorithms such as AODV and flooding. The references are valid, although some more current ones should be included. The text is well written and well structured, although there are some aspects that should be better explained and detailed below.

1. Pag. 1. Introduction. Some reference to the terms upward and downgard.
2. Pag. 2. Line 80. The terms ADO and DOAG are not defined.
3. Pag.2 Line 81. How would the concepts of root, child, etc, be defined in this context?
4. Pag. 4. Line 157. How is the rank calculated?
5. Page 4. Line 159. How are the metrics obtained?
6. Page 5. Line 184. How is it done?
7. Section 3.3. Is this a mechanism similar to that of AODV v2?
8. PAg 5. Line 203. A reference to the simulator is missing.
9. Pag. 6. Line209. Would not it be upward instead of upstream?
10. Pag 6. Line 216. Include references to AODV and flooding.
11. Section 4. A new protocol that is more used in WSN and not in AODV could be included in the comparisons. For example, some routing protocol based on energy.

Author Response

We would like to thank the reviewer for their work. Our answers to each of their comments and requests are listed in the attached file.

Round 2

Reviewer 1 Report

The authors have properly addressed of the concerns raised in the previous review.

Reviewer 2 Report

The authors have made all the suggested changes.